A new ornithurine from the Early Cretaceous of China sheds light on the evolution of early ecological and cranial diversity in birds

Huang Jiandong 1
Wang Xia 2 wangxia8383@gmail.com
Hu Yuanchao 1
Liu Jia 1
Peteya Jennifer A. 3
Clarke Julia A. 2 julia_clarke@jsg.utexas.edu
1 Department of Research, Anhui Geological Museum , Anhui , China
2 Department of Geological Sciences, Jackson School of Geoscience, University of Texas at Austin , Austin, TX , United States
3 Department of Biology & Integrated BioScience Program, University of Akron , Akron, OH , United States
Hutchinson John
Electronic publication date: 2016 Mar 15
Publication date: 2016
Volume: 4
Electronic Location ID: e1765
Received 2015 Jun 19; Accepted 2016 Feb 15
Copyright: ©2016 Huang et al.
Copyright year: 2016
Copyright holder: Huang et al.
License: This is an open access article distributed under the terms of the Creative Commons Attribution License, which permits unrestricted use, distribution, reproduction and adaptation in any medium and for any purpose provided that it is properly attributed. For attribution, the original author(s), title, publication source (PeerJ) and either DOI or URL of the article must be cited.
License URL: https://creativecommons.org/licenses/by/4.0/

Keywords: Avialae, Fossil, Ontogeny, Ornithurae, Jehol Biota

Funding: Special Scientific Research Fund of the Non-Profit Sector Project of China 201511054 US National Science Foundation NSF EAR 1251922 This research is supported by the Special Scientific Research Fund of the Non-Profit Sector Project of China (Grant No 201511054) and the US National Science Foundation (NSF EAR 1251922 to J.A.C.). The funders had no role in study design, data collection and analysis, decision to publish, or preparation of the manuscript.

==============================
Despite the increasing number of exceptional feathered fossils discovered in the Late Jurassic and Cretaceous of northeastern China, representatives of Ornithurae, a clade that includes comparatively-close relatives of crown clade Aves (extant birds) and that clade, are still comparatively rare. Here, we report a new ornithurine species Changzuiornis ahgmi from the Early Cretaceous Jiufotang Formation. The new species shows an extremely elongate rostrum so far unknown in basal ornithurines and changes our understanding of the evolution of aspects of extant avian ecology and cranial evolution. Most of this elongate rostrum in Changzuiornis ahgmi is made up of maxilla, a characteristic not present in the avian crown clade in which most of the rostrum and nearly the entire facial margin is made up by premaxilla. The only other avialans known to exhibit an elongate rostrum with the facial margin comprised primarily of maxilla are derived ornithurines previously placed phylogenetically as among the closest outgroups to the avian crown clade as well as one derived enantiornithine clade. We find that, consistent with a proposed developmental shift in cranial ontogeny late in avialan evolution, this elongate rostrum is achieved through elongation of the maxilla while the premaxilla remains only a small part of rostral length. Thus, only in Late Cretaceous ornithurine taxa does the premaxilla begin to play a larger role. The rostral and postcranial proportions of Changzuiornis suggest an ecology not previously reported in Ornithurae; the only other species with an elongate rostrum are two marine Late Cretacous taxa interpreted as showing a derived picivorous diet.

Introduction

So far more than 16 species of ornithurine birds have been reported from an array of localities of the Early Cretaceous of northern China, Jehol Biota (Fig. 1) Only six are (i.e., Yanornis martini, Gansus yumenensis, Hongshanornis longicresta, Archaeorhynchus spathula, Iteravis huchzermeyeri and Gansus zheni) represented by multiple specimens. The majority of taxa are known from the Jiufotang Formation (122.1 ± 0.3 Ma, Chang et al., 2009) of Western Liaoning Province, China with Archaeorhynchus is known from both the older Yixian Formation as well as the Jiufotang Formation. Archaeornithura meemannae is the oldest taxon in the Hongshanornithidae; it is known from the Huajiying Formation (Wang et al., 2015). Most Chinese ornithurines have been proposed to be volant and semi-aquatic, with Gansus yumenensis, known from the Xiagou Formation (early Aptian, Suarez et al., 2013), proposed to likely represent a foot-propelled diver (You et al., 2006; Nudds et al., 2013). Some more basal taxa placed outside of Ornithurae (e.g., Jianchangornis microdonta and Archaeorhynchus spathula), have been proposed to have occupied fully terrestrial niches (Zhou, Zhang & Li, 2009; Zhou, Zhou & O’Connor, 2013). However, despite the ever increasing number of avialan specimens discovered from China, Early Cretaceous birds were found to be substantially impoverished in ecology (Mitchell & Makovicky, 2014).

Figure 1 Distribution of Ornithurae birds from Early Cretaceous of China.

So far more than 16 species of ornithurine birds have been reported from multiple locations of the Early Cretaceous of northern China, Jehol Biota (i.e., Songlingornis linghensis, Chaoyangia beishanensis, Yanornis martini, Yixianornis grabaui, Gansus yumenensis, Hongshanornis longicresta, Archaeorhynchus spathula, Longicrusavis houi, Parahongshanornis chaoyangensis, Tianyuornis cheni, Archaeornithura meemannae, Jianchangornis microdonta, Schizooura lii, Piscivoravis lii, Iteravis huchzermeyeri, Gansus zheni, Xinghaiornis lini; Hou, 1997; Zhou & Zhang, 2001; Zhou & Zhang, 2005; Clarke, Zhou & Zhang, 2006; You et al., 2006; Zhou & Zhang, 2006; Zhou, Zhang & Li, 2009; O’Connor, Gao & Chiappe, 2010; Zhou, Zhou & O’Connor, 2012; Zhou, Zhou & O’Connor, 2013; Wang et al., 2013; Chiappe et al., 2014; Liu et al., 2014; Zheng et al., 2014; Zhou, O’Connor & Wang, 2014a; Zhou, Zhou & O’Connor, 2014b; Wang et al., 2015).

Known diversity in rostral shape of these Jehol taxa has also been limited; the majority have relatively short rostra and the only other birds with an elongate rostrum proposed to be part of Ornithurae are Xinghaiornis lini, Juehuaornis zhangi and Dingavis longimaxilla (Wang et al., 2013; Wang, Wang & Hu, 2015; O’Connor, Wang & Hu, 2016). Here, we describe and evaluate the phylogenetic position of a new ornithurine species with an elongate rostrum and gastroliths from a relatively new locality (Sihedang), Early Cretaceous, Jiufotang Formation of Liaoning Province. This species contributes to our understanding of Mesozoic avialan cranial diversity and evolution.

The electronic version of this article in Portable Document Format (PDF) will represent a published work according to the International Commission on Zoological Nomenclature (ICZN), and hence the new names contained in the electronic version are effectively published under that Code from the electronic edition alone. This published work and the nomenclatural acts it contains have been registered in ZooBank, the online registration system for the ICZN. The ZooBank LSIDs (Life Science Identifiers) can be resolved and the associated information viewed through any standard web browser by appending the LSID to the prefix “http://zoobank.org/”. The LSID for this publication is: urn:lsid:zoobank.org:pub:DE31419A-CFC4-40BC-A6C9-91B0FED3D967. The online version of this work is archived and available from the following digital repositories: PeerJ, PubMed Central and CLOCKSS.

Systematic Paleontology

Aves Linnaeus, 1758	
Ornithurae Haeckel, 1866 sensu Gauthier & De Queiroz, 2001	
Changzuiornis ahgmi gen. et sp. nov.	

Holotype specimen. A nearly-complete skeleton with feather impressions (Fig. 2; AGB5840; “AGB” refers to the Anhui Gushengwu Bowugan in pinyin, or Anhui Paleontological Museum, which is the Anhui Geological Museum). The skeleton is preserved primarily in lateral view. Parts of the pectoral and pelvic girdles are partially disarticulated as are some of the manual phalanges and caudal vertebrae.

Figure 2 Photograph of the Holotype Changzuiornis ahgm (AGB5840).

Anatomical abbreviations: co, coracoid; cv, cervical vertebra; f, feathers; fe, femur; fu, furcula; ga, gastrolith; hu, humerus; il, ilium; ins, internarial septum; ios, interorbital septum; mcI–III, metacarpals I–III; pd I–IV, pedal digits I–IV; phI-1, the first phalanx of digit I; phI-2, the second phalanx of digit I; phII-1,the first phalanx of digit II; phII-2, the second phalanx of digit II; pu, pubis; py, pygostyle; ra, radius; rad, radiale; ri, rib; sc, scapula; sk, skull; ti, tibiotarsus; tm, tarsometatarsus; tv, thoracic vertebra; ul, ulna; uln, ulnare. Numbers (1, 2) show the locations of SEM imaging of feather remains. Insets show melanosome morphologies from the two sample locations.

Locality and horizon. Sihedang locality, Lingyuan City, western Liaoning Province, China. Jiufotang Formation, Early Cretaceous (Aptian; Chang et al., 2009).

Etymology: The genus name derives from Chinese pinyin “Changzui” referencing the long beak and Greek word “ornis-” for bird; and the species name refers to Anhui Geological Museum (AHGM) where the holotype specimen is housed.

Diagnosis

The placement of Changzuiornis ahgmi within the clade Ornithurae is supported by seven unambiguously optimized synapomorphies (listed in the Phylogenetic Analysis section below). It diagnosed by a combination of morphologies not seen in other described ornithurines: The rostrum is elongate, comprising greater than 60% of the total skull length. Most of this elongate rostrum is made up of maxilla, a characteristic not present in the avian crown clade in which most of the rostrum and nearly the entire facial margin is made up by premaxilla. The only other avialans known to exhibit an elongate rostrum with the facial margin comprised primarily of maxilla are Xinghaiornis, Juehuaornis and Dingavis and derived Late Cretaceous ornithurines previously placed phylogenetically as among the closest outgroups to the avian crown clade (i.e., Ichthyornis, hesperornithine taxa). It differentiated from Xinghaiornis by its much smaller size, many tiny teeth on the lower jaw, U-shaped furcula, metacarpal III sub-equal to metacarpal II in distal extent, a carpometacarpus with both proximal and distal fusion, and a tarsometatarsus that is completely fused. The skull is about 15% longer in Changzuiornis and Juehuaornis compared to Dingavis (Table S1). In addition, the scapula is proportionally longer in Changzuiornis, Juehuaornis and Dingavis have a manual phalanx II:2 shorter than manual phalanx II:1 while in Changzuiornis this phalanx is longer (Table S1). The new species is differentiated from Hesperornithes and Ichthyornis by significantly smaller teeth with less recurved crowns, the presence of a distinct dorsal process or “forking” of the posterior dentary, and the presence of a pubic symphysis. It is additionally differentiated from Ichthyornis by robust and more abbreviate furcular rami, a narrower sternal margin of the coracoid, and a significantly more elongate scapular acromion. While if new data shows that Xinghaiornis, Juehuaornis and Dingavis form a clade that constitutes the same genus, the genus name Juehuaornis would have priority for Dingavis longimaxilla and Changzuiornis ahgmi.

Description

Skull

The skull (Fig. 3) is preserved in right lateral view. The rostrum is elongate (48 mm), comprising ∼68% of the total skull length. It is longer than those reported in the only other long-rostrum Jehol taxa, the longipterygid enantironithines, Longipteryx (64%), Rapaxavis pani (65%), Longirostravis (60–64%) and Shanweiniao (62%), and comparable in proportions to those of extant woodcocks. The dorsal processes of the premaxillae are not fused to each other posteriorly (Fig. 4). While the right dorsal process is missing, the left one visibly extends posteriorly to contact the frontal. The posterior tip of the facial margin of the left premaxilla is missing while the articulating tip of the maxilla is visible and sharply tapered rostrally (Fig. 4). From the length of the exposed maxilla, the premaxilla comprised less than one half of the facial margin (rostrum), a condition also seen in enantiornithines with long rostra (e.g., Longirostravis and Shanweiniao; Hou et al., 2004; O’Connor et al., 2009).

Figure 3 Skull of Changzuiornis ahgm.

Anatomical abbreviations: bh, basihyal; ce, ceratobranchial; de, dentary; fp, frontal process; fpc, frontals and premaxillae contacting area; fr, frontal; ins, internarial septum; la, lacrimal; ma, maxilla; na, nasal; pa, parietal; pd, predentary; pm, premaxillae; q, quadrate. Inset showing the tiny teeth preserved on dentary.

Figure 4 Close-up of the skull of Changzuiornis ahgm.

Anatomical abbreviations: atm, articulating tip of the maxilla; dpm, dorsal process of the maxilla; en, external nares; fd, forked dentary; fp, frontal process; fpc, frontals and premaxillae contacting area; ins, internarial septum; ios, interorbital septum; lfp, left frontal process; so, scleral ossicles.

The right nasal is laterally exposed with a descending process that lies adjacent to a small dorsal process of the maxilla (Fig. 4). The nasals appear not to have contacted along the dorsal midline but were likely separated by the frontal processes of the premaxillae as seen in Confuciusornis (Chiappe et al., 1999; Fig. 3) or underlay parts of these processes. However, the right nasal does not appear preserved in life position. The external nares are elongate and relatively narrow. A thin vertical sheet-like element visible in the narial region extends the length of the external nares and is interpreted as an internarial septum (Figs. 3 and 4). The preservation of an internarial septum in Changzuiornis is the first known occurrence in a Mesozoic bird. A similar sheet like element is visible below several the preserved scleral ossicles, consistent with at least a partial interorbital septum formed by the mesethmoid; Fig. 4). A mesethmoid is present in an array of basal avialans including Confucisuornis and ornithurines such as Yixianornis as well as Enantiornithes (e.g., Longipteryx haoyangensis and Schizooura lii; Zhou & Zhang, 2001; Clarke, Zhou & Zhang, 2006; Zhou, Zhou & O’Connor, 2012).

Parts of the rostral jugal and posterior-most quadratojugal are preserved. Just dorsal to the small rostral portion of the jugal (preserved at its contact with the maxilla), the poorly preserved remnants of the right lacrimal are visible (Fig. 3). The quadrate is poorly preserved but shows an arcuate posterior margin and a relatively elongate orbital process (Fig. 3). The frontoparietal suture appears to have been open (Fig. 3). The posterior portion of the skull is severely crushed. A small roughly t-shaped element preserved nearly the posterior margin of the orbit could represent a remnant of a postorbital; however, this cannot be ascertained with confidence and a postorbital is currently unknown in Ornithurae (Martin, 2011).

The mandibles are relatively straight and taper anteriorly. A predentary bone lies in front of left mandibular ramus (Fig. 3). Such a structure has also been reported in Hongshanornis. Ichthyornis, Hesperornis, and Parahesperornis and an array of other ornithurine birds (e.g., Zhou & Zhang, 2005). The dentary is forked posteriorly whereas the posterior dentary lacks a dorsal process in Ichthyornis and Hesperornithes (Fig. 4). At least three large mental foramina are visible in a shallow groove near the dorsal margin of the rostral dentary. The mandible is obscured by partly crushing posterior to the dentary.

Both premaxillae are completely edentulous, unlike those in the enantiornithines Longipteryx, Rapaxavis, Longirostravis and Shanweiniao (O’Connor & Chiappe, 2011). Approximately seven tiny, narrow, pointed and unserrated loose teeth are exposed closely between the maxilla and dentary (Fig. 3). One appears to be in situ, indicating an association with the dentary. No teeth seem preserved on the tip of dentary. Small neurovascular foramina are visible on the dorsal surface of the premaxilla consistent with the presence of a rhamphotheca (e.g., Chiappe et al., 1999).

Parts of the hyoid elements are preserved. Two recurved ceratobranchials are preserved with breakage separating their rostral-most tips from the rest of their preserved length. These rostral tips lie close to the right mandible (Fig. 3). A small, isolated element preserved below their tips may comprise a very small, ossified basihyal (Fig. 3: bh).

Vertebral column

The entire cervical series is preserved. The anterior cervicals are visibly heteroceolic while posterior-most cervicals do not appear heterocoelic, but are poorly exposed. The mid-series (fifth–eighth) cervical vertebrae are elongate. Elongate costal processes are present on the second through eighth vertebrae of the eleven or twelve vertebrae in the series (Fig. 2). The thoracic series is obscured by the right scapula. However, approximately ten thoracic vertebrae are discernable with large lateral fenestrae (Fig. 2) and amphiplatyan centra articulations. The neural spines are craniocaudally elongate. The sacrum is badly crushed, and no morphologies can be discerned. Only the approximate outlines of several free caudal vertebrae are visible while the pygostyle is clearly exposed. It is strongly mediolaterally compressed and short, approximately two anterior caudal vertebrate in length (Fig. 2).

Pectoral girdle and limb

The right coracoid is exposed in ventral view (Fig. 2). The left coracoid is obscured by the furcula and the right humerus. A large flange-like procoracoid process and an abbreviated lateral process are present (Fig. 5). The furcula is U-shaped and relatively thin; a furcular apophysis appears to be absent. The omal tips of the rami are severely crushed. The left scapula, lying under the furcula and several ribs, preserves an elongate and hook-shaped acromion process (Fig. 5). The posterior ends of both scapulae are missing. However, their preserved length approaches that of the humeri. The sternum is not visible. Approximately five large gastroliths (∼30 mm in diameter; Fig. 2) are visible in the abdominal region, between the coracoid and the posterior thoracic vertebrae. A small quantity of fine grit is intermixed with these larger stones. Unfortunately, a textured depression just posterior to the five preserved stone may indicate a larger quantity of such stones were present but lost during collection or early preparation.

Figure 5 Pectoral girdle and forelimb of Changzuiornis ahgm.

Anatomical abbreviations: ap, acromion process; co, coracoid; dc, deltopectoral crest; fu, furcula; h, head; hu, humerus; lp, lateral process; ri, rib; sc, scapula.

Both humeri are exposed in anterior view (Figs. 2 and 5), and are just slightly shorter than the ulnae. The dorsally-directed deltopectoral crest is recurved and projected approximately equal to shaft width (Fig. 5). It extends distally for just greater than 1/3 of the total humeral length and then grades gently into the shaft. The head is weakly globose, and a small m. acrocoracohumeralis ligament scar is visible. On the distal humerus, the dorsal condyle is more elongate than the ventral. The radii are narrower than the slightly-bowed ulnae. Few morphologies of the proximal or distal radii and ulnae are visible. The radiale is larger than the ulnare, and the ulnare appears differentiated into distinct dorsal and ventral rami.

The right carpometacarpus is exposed in dorsal view and the left, in ventral view. The metacarpals are fused proximally and distally. Metacarpal III is significantly narrower than metacarpal II. It is straight and closely appressed to metacarpal II. On the left carpometacarpus, a small piciform process is visible (Fig. 6). Metacarpal I bears a very weakly-projected extensor process (Fig. 6); its anterior margin is nearly straight. Digit I:1 extends over the ½ length of metacarpus and bears a small claw. Manual digit II:1 is approximately the same length as digit II:2. The posterior margin of digit II:1 is strongly compressed dorsoventrally, and digit II:2 shows a distinct fossa that in crown clade Aves is the attachment site for the leading edge primary feather (Fig. 6; Hieronymus, 2015). An impression of a small digit II ungual is preserved on the left side. It is only weakly recurved. Phalanx III:1 is not visible.

Figure 6 Carpometacarpus of Changzuiornis ahgm.

Anatomical abbreviations: ep, extensor process; fo, fossa; im, impression; mcI–III, metacarpals I–III; phI-1, the first phalanx of digit I; phI-2, the second phalanx of digit I; phII-1,the first phalanx of digit II; phII-2,the second phalanx of digit II; phII-3,the third phalanx of digit II; pip, piciform process; ra, radius; ul, ulna.

Pelvic girdle and limb

Most of the partially-disarticulated pelvic girdle is obscured by crushing. The rod-like pubes (Figs. 2 and 7) have separated from the rest of the pelvic elements and preserve a relatively elongate symphysis. Their distal ends are not visibly expanded. The femur is notably shorter than the tarsometatarsus (Table 1) and its shaft is straight. The trochanteric crest is weakly projected proximally. The attachment of the capital ligament is indicated by a distinct notch on the lateral surface of the proximal femur (Fig. 7). The left tibia preserves a slightly proximally-projected anterior cnemial crest (Fig. 7). The morphology of the distal condyles is poorly preserved, but they appear fused to the tibia and are visibly separated by an intercondylar incisure (Fig. 7). The distal tarsals are fused to the metatarsals.

Figure 7 Pelvic girdle of Changzuiornis ahgm.

Anatomical abbreviations: cc, cnemial crest; fcl, fovea for capital ligament; fe, femur; pu, pubis; sy, symphysis; tc, trochanteric crest; tct, tibialis cranialis tubercle; ti, tibiotarsus; tm, tarsometatarsus.

Table 1 Measurements of the new specimens referred to Changzuiornis angmi (AGB5840) (in cm) L/R.

Premaxilla length along facial margin	3.22	Phalanx II:3	0.36	
Dentary length, total	5.48	Phalanx III:1	–	
Rostrum length (from the tip of premaxilla to frontal and premaxilla contact)	4.80	Ilium length total	3.04	
Dentary dorsoventral height at anterior tip	0.11	Ilium preacetabular	1.43	
Cervical vertebra average length	0.71	Ischium length	3.84a	
Sacrum length	3.3a	Pubis length	3.43	
Sternum length on midline	–	Pubis average shaft diameter	0.21	
Scapula maximum length	4.64R	Pubis symphysis length	1.35	
Coracoid height	2.38R	Pelvic limb		
Coracoid sternal margin length	–	Femur maximum length	2.94R	
Coracoidal lateral process length	1.19R	midshaft width	0.35	
Furcula: length clavicular ramus	1.67	Tibia maximum length, not including cnemial crest	5.37/5.40	
Humerus maximum length	5.04/5.01	Tarsometatarsus maximum length	3.60/3.61	
Radius length	5.06/4.86	Pedal phalanx I:1 length	0.72R	
Radius midshaft width	0.25/0.23	Pedal phalanx II:1	0.95/0.87	
Ulna length	5.2/5.36	Pedal phalanx II:2	1.07/0.99	
Ulna midshaft width	0.4/0.36	Pedal phalanx III:1	0.89/0.71	
Carpometacarpus maximum length	3.01/3.13	Pedal phalanx III:2	0.84/0.68	
Metacarpal I length	0.54/0.51	Pedal phalanx III:3	0.59a/0.63	
Metacarpal III width	0.14/0.14	Pedal phalanx IV:1	0.77/0.74	
Metacarpal II width	0.22/0.32	Pedal phalanx IV:2	0.72/0.71	
Manual phalanx I:1	1.17	Pedal phalanx IV:3	0.54/0.60	
Manual Phalanx I:2	0.54R	Pedal phalanx IV:4	0.54/0.65	
Manual Phalanx II:1	1.29/1.16	Remiges: maximum length distal primaries (9 & 8?)	15.36	
Manual Phalanx II:2	1.44/1.26			
Notes.

a Estimated.

L Left

R Right

The tarsometatarsus is mediolaterally compressed. A prominent midline m. tibialis cranialis tubercle is visible on the proximodorsal left tarsometatarsus (Fig. 7). The j-shaped metatarsal I is well-exposed in articulation with a short hallux; pedal digit I:1 is longer than the small ungual, I:2. Pedal phalanges are preserved in association with both tarsometatarsi. They are narrow with deep flexor pits and only weakly recurved unguals. In all digits (Fig. 2), the unguals are shorter than their penultimate phalanges. The flexor tubercles on the unguals are proximally located, unlike the more distally located tubercles used to infer the presence of webbing in Gansus yumenensis (You et al., 2006). In digit II, the second phalanx slightly exceeds the first in length. Both non-terminal phalanges of this digit are the longest of the pedal digits. In digit three, the penultimate phalanx is slightly shorter than the more proximal phalanges. By contrast, in the fourth digit, the penultimate phalanx is slightly longer than the more proximal phalanges.

Feathers

Feather remains are poorly preserved (Fig. 2). Remnants of body contour feathers are associated with the posterior cranium and the cervical area as well as near wing and leg elements. In addition, several primary feathers are associated with both forelimbs. Traces of rachis and barbs are distinguishable only in certain distal regions of these feathers, which are otherwise preserved as grey-white impressions. One primary feather associated with left manual digit II:1 is approximately 106 mm in length. The rachis of this asymmetrically veined feather is discernable. The lengths of the rest of the preserved remiges are impossible to determine with confidence. Scanning electron microscopy (SEM) results show that wing and leg/tail feather samples contain melanosome molds that are highly aligned, closely spaced and elongate in shape (Fig. 2). Their aspect ratio (length:width ratio:2.29–4.99) is typical of eumelanosomes seen in black feathers (Vinther et al., 2009; Li et al., 2010; Clarke et al., 2010).

Maturity

The following anatomical features present the holotype specimen have been proposed to indicate an adult at death (Forster et al., 1998; Xu & Norell, 2004; Turner et al., 2007; Gao et al., 2012; Godefroit et al., 2013): (i) the texture of the bones is regular and well ossified bearing articular facets and muscular scars (e.g., proximal and distal ends of the coracoid, the humerus, the ulna, the pubis and the femur); (ii) the frontals are fused to each other; (iii) cervical ribs and cervical vertebrae co-ossified to enclose transverse foramina; (iv) distal carpals coosified with metacarpals II and III; (v) metatarsals II, III and IV fused throughout their length.

Phylogenetic Anlysis

We investigated the phylogenetic position of Changzuiornis using a dataset of 220 morphological characters modified from that of Li et al. (2014). This dataset (Appendixces S1 and S2) was revised by modifying and ordering one character (relative length of pedal digits; character 217; Appendix S1) and adding Gansus yumenensis. All analyses were performed using PAUP 4.0b10 (Swofford, 2003). Heuristic searches were used given the size of the taxonomic sample (40 ingroup and outgroup taxa). Three thousand replicates of random stepwise addition (branch swapping: tree-bisection-reconnection) were performed holding only one tree at each step. Branches were collapsed to create polytomies if the minimum branch length was equal to zero. One thousand bootstrap replicates with ten random stepwise addition heuristic searches per replicate were also performed with the same settings as in the primary analysis. Bootstrap support for those nodes recovered in greater than 50 percent of the 500 replicates performed and Bremer support values are reported to the right of the node to which they apply (Format: Bootstrap/Bremer in Fig. 8). Bremer support values were calculated by iterative searches for suboptimal trees in PAUP 4.0b10 using the same heuristic search strategy as the primary analysis. 15 most parsimonious trees (MPTs) were recovered (L = 585, CI = 0.50, RI = 0.79, RC = 0.40; PIC only).

Figure 8 Strict consensus cladogram illustrating the phylogenetic position of Changzuiornis ahgm.

(length L: 585, CI: 0.50, RI 0.79, RC 0.40 (PIC only)). Bootstrap support for those nodes recovered in greater than 50% of the 1,000 replicates performed and Bühler, Martin & Witmer (1988) support values are reported to the right of the node to which they apply (Format: Bootstrap/Bremer). Skulls are illustrated to show the change of facial margin composition along the evolution of avialans. Red, maxilla; yellow, premaxilla. The skull of Rapaxavis (Longipterygidae) with elongated rostrum is also shown here.

The strict consensus tree (Fig. 8) recovers Changzuiornis ahgmi within Ornithurae closer to Aves than the clade formed by Yixianornis, Songlingornis and Yanornis but basal to Gansus yumenensis and the clade Gansus zheni + Iteravis. The later clade is supported by one unambiguous synapomorphy, a reversal (170:0, distal end of pubes expanded or flared). The placement of Changzuiornis ahgmi within Ornithurae is supported by seven unambiguously optimized synapomorphies (numbers refer to characters and states listed in Appendix S2): 9:1, nasal process of premaxilla long, closely approaching frontal; 87:1, coracoid, procoracoid process present; 140:3, semilunate carpal and metacarpals complete proximal and distal fusion; 142:1, metacarpal III, anteroposterior diameter as a percent of same dimension of metacarpal II less than 50%; 151:1, manual digit II, phalanx 1 strongly dorsoventrally compressed, flat caudal surface; 192:1, metatarsal III proximally displaced plantarly, relative to metatarsals II and IV; 218:1, hallux, claw to phalanx proportions, 1:1, shorter. Unlike previous studies (e.g., Clarke & Norell, 2002; Clarke, 2004; Clarke, Zhou & Zhang, 2006; Zhou, Clarke & Zhang, 2008; O’Connor, Gao & Chiappe, 2010), Apsaravis is placed more closely to Aves than Ichthyornis, which is supported by six unambiguous synapomorphies (4:1, dentary teeth absent; 6:1, dentaries joined by a bony symphysis; 52:2, cervical vertebrae with heterocoelous anterior and posterior centra; 204:1, metatarsal II shorter than metatarsal IV, but reaching distally farther than the base of the metatarsal IV trochlea; 212:1, over half of the glenoid facet lies omal to the cotyla in lateral view). The skull characters (especially characters of the dentition) are missing data in Gansus yumenensis, which may influence this optimization. Indeed, when we remove Gansus yumenensis from the phylogenetic analysis, Apsaravis is placed as basal to Ichthyornis + Hesperornithes as in all previous analyses (e.g., Clarke, 2004; Clarke, Zhou & Zhang, 2006; Zhou, Clarke & Zhang, 2008; O’Connor, Gao & Chiappe, 2010; Zhou, Zhou & O’Connor, 2012; Li et al., 2014). Apsaravis shares loss of teeth with Aves, a highly homoplastic character in Avialae, but lacks features seen in Ichthyornis and Aves that have a high consistency index including a hypotarsus with grooves and ridges and a medial coracoidal margin that is flat to convex rather than concave with a midline groove otherwise seen in Enantiornithes and more basal avialans (Clarke & Norell, 2002; Clarke, 2004). Hesperornithes, Ichthyornis and Aves also show a deep medial extensor groove on the tibiotarsus absent in Apsaravis (Clarke & Norell, 2002) and an array of other derived states. What is clear is that with the discovery of further diversity in Ornithurae, known homoplasy is also increasing similar to the situation in basal paravian relationships (e.g., Xu et al., 2011).

Discussion

Jehol Biota continues to reveal important new data on the evolution of morphology in the transition from bipedial terrestrial dinosaurs to volant forms. While character systems tied to locomotor mode have received the most scrutiny, some recent work has begun to address what other changes in ecology and morphology shift as part with this transition, including the gross morphology of the brain (e.g., Balanoff, Bever & Norell, 2014; Wang, Wang & Hu, 2015; O’Connor, Wang & Hu, 2016) and cranial shape and ontogeny (Bhullar et al., 2012).

The presence of feeding ecologies novel for Dinosauria may be expected to be seen within Avialae enabled by evolution of their novel locomotor mode. However, a recent study by using skeletal morphology to predict ecology in both living and extinct birds proposed that the ecological diversity of Cretaceous birds is anomalously low in the Jehol ecosystem and dominated by ground-foraging granivores/insectivores, similar to sparrows or pigeons (Mitchell & Makovicky, 2014). While within living birds, there is enormous diversity in skull shape so far in Mesozoic taxa most diversity has been in dentition (Li et al., 2014), which could be considered one explanation for the low estimates of ecological diversity. Dentition shows complex trends in Avialae including patterns of loss or reduction and evolution of dental morphologies not present in more basal dinosaurs. Changzuiornis adds to this known diversity in dentition. The teeth in Changzuiornis apprear limited to the more posterior portion of dentary, a condition not seen in other birds. Evolution of tooth loss in this clade is proposed to start in the rostral-most part of the premaxillae (Clarke, Zhou & Zhang, 2006; Louchart & Viriot, 2011). However, there is significant diversity making optimization of ancestral traits largely ambiguous; different dental patterns are distributed as follows: (1) edentulous premaxillary tip (e.g., Yanornis, Yixianornis); (2) edentulous premaxilla (e.g., Hesperornis, Ichthyornis, Gansus zheni, Iteravis, and Hongshanornis); (3) edentulous upper jaw (e.g., Jianchangornis); (4) edentulous upper jaw and rostral dentary (e.g., Changzuiornis); (5) fully edentulous (e.g., Apsaravis, Archaeorhynchus, Schizooura).

In marked contrast with dentition, so far known diversity in rostral shape in Jehol taxa has been limited (Li et al., 2014; Wang, Wang & Hu, 2015; O’Connor, Wang & Hu, 2016); Five distinct taxa within ornithurine birds (i.e., Changzuiornis, Juehuaornis, Dingavis, Hesperornithes, Ichthyornis) show elongate rostri along with one enantiornithurine clade (Longipteryidae; Fig. 4). This rostral shape is rare even in more basal maniraptoran dinosaurs. A developmental explanation of some of these data has been proposed (Bhullar et al., 2012; Bhullar et al., 2015). Mesozoic birds have plesiomorphic rostral morphologies not seen in extant birds including premaxillae restricted to the tip or cranial half of the facial margin. Recent studies propose peramorphosis in formation of the distinctive elongate avian beak (comprised primarily of premaxilla) phylogenetically between Confuciusornis and Yixianornis. This shift was hypothesized to be linked to the evolution of the beak into a precise grasping mechanism following increasing specialization of the forelimbs for flight (Bhullar et al., 2012). Supporting the idea of a developmental constraint persisting into basal Ornithurae on rostral development, all Mesozoic taxa with elongate rostra, including Changzuiornis, Dingavis accomplish this elongation through elongation of the maxilla and not the premaxilla (Fig. 4; O’Connor, Wang & Hu, 2016). Given the age and phylogenetic position of the ornithurines that show this morphology, the proposed change in development must occur in the Late Cretaceous, close to the timing of origin for the radiation of all extant birds. The fact that the elongate maxillae were not present in more derived taxa implies that in Aves this skull configuration provided less structural stability (O’Connor, Wang & Hu, 2016).

The structure and function of the extant avian rostrum in grasping is closely tied to cranial kinesis or zones of mobility or flexure (Bock, 1964; Zusi, 1984). Data from Changzuiornis may inform our spotty understanding of this transition. Small vascular foramina (nutrition foramina) are visible on the dorsal surface of the premaxilla, indicating that the anterior margin of the upper jaw was covered by a keratinous sheath or bill. However, as indicated by the comparatively broad nasals, the position of the holorhinal nostril (terminating anterior to the premaxilla/frontal contact), and the loss of teeth in the upper jaw, we think rhynchokinesis, the significant proximal and/or distal rostral flexure seen in many extant birds (e.g., cranes, rails, shorebirds, swifts and hummingbirds) may not have been possible (Bock, 1964; Zusi, 1984; Bout & Zweers, 2001; Estrella & Masero, 2007).

Prokinesis, in which the upper jaw pivots only at a narrow and well defined craniofacial hinge is widespread in extant birds including galloanserines and has been considered primitive for at least the group including Hesperornithiformes, Ichthyornis and Aves given the morphology of the premaxillae-nasofrontal contact in those taxa (Bühler, Martin & Witmer, 1988; Witmer, 1995). By contrast, the skull of Confuciusornis has been estimated to be only marginally kinetic or akinetic; it exhibits a relatively indistinct naso-frontal zone located approximately on the rostral edge of the cranial vault above part of the orbit (Chiappe et al., 1999; Fig. 4). Unlike Confuciusornis, but similar to Ichthyornis, Hesperornis and extant prokinetic taxa, in Changzuiornis the area of contact between the frontals and premaxillae lies anterior to the orbit and is demarcated by zone of dorsal concavity (Fig. 4). The development of an internarial septum in the new species could intuitively limit any more rostral bending (Zusi, 1984). However, the presence of an internarial septum in extant Palaeognthae which exhibit an apparently derived form of rynchokinesis problematizes any simple inference for or against further more rostral flexure (Zusi, 1984; Gussekloo & Bout, 2005). We propose that a form of kinesis may date to at least the common ancestor of Changzuiornis and Aves. However we caution that the rostrum of all of these basal taxa is not comprised of only a single bone element but has a prominent juncture in the middle of the facial margin between the small premaxilla and large maxilla that may be expected to affect bending. If Hesperornithes and Ichthyornis show a form of prokinesis, as previously proposed, then kinetic demands related to increased grasping function may predate or potentially drive the developmental shift observed.

Conclusions

With its combination of a long and slender bill, proximally-projected cnemial crests, relative long tibiotarsus, and small pedal unguals, Changzuiornis represents a distinct departure from other known Jehol birds and contributes to our understanding of morphological and ecological diversity. That highly specialized species usually occur in low numbers (Julliard, Jiguet & Couvet, 2004; Şekercioğlu, Daily & Ehrlich, 2004) in a given fauna, maybe one explanation for comparative paucity of previously known diversity in rostral morphology in the Jehol Biota. At the same time, Mesozoic avialans maybe be subject to developmental constraints on rostral shape (Bhullar et al., 2012) and most diversity in feeding ecology is reflected in diversity in dentition. If so, inferences of feeding ecology primarily from rostral shape in Mesozoic birds should be approached with caution (Li et al., 2014). Changzuiornis constrains further when this shift in rostral development (Bhullar et al., 2012; Bhullar et al., 2015) may have occurred. So far it appears to be a Late Cretaceous phenomenon that arose close to the timing of crown clade origin and as such should be further investigated as a possible key innovation in their remarkable radiation. Increased kinesis, possibly as a form of prokinesis, is estimated to predate this developmental shift.

Supplemental Information

Appendix S1 Character list

Click here for additional data file.

Appendix S2 Character matrix

Click here for additional data file.

Table S1 Measurements of Changzuiornis angmi, Juehuaornis zhangi and Dingavis longimaxilla (in mm)

Click here for additional data file.

We thank Zhiheng Li for help with the data matrix and Anjan Bhullar, Arhat Abzhanov, and Zack Morris for discussion. The comments of two anonymous referees improved the manuscript.

Additional Information and Declarations

Competing Interests

Author Contributions

Data Availability

New Species Registration

Julia A. Clarke serves as an academic editor for PeerJ.

Jiandong Huang, Yuanchao Hu and Jia Liu contributed reagents/materials/analysis tools.

Xia Wang and Julia A. Clarke conceived and designed the experiments, performed the experiments, analyzed the data, wrote the paper, prepared figures and/or tables, reviewed drafts of the paper.

Jennifer A. Peteya analyzed the data.

The following information was supplied regarding data availability:

Some raw data (measurements) has been provided in Table S1 and the Supplemental Information.

The following information was supplied regarding the registration of a newly described species:

Name of the new taxon: Changzuiornis ahgmi sp.nov.

LSID:urn:lsid:zoobank.org:pub:DE31419A-CFC4-40BC-A6C9-91B0FED3D967.

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
