# Peer review of "A new ornithurine from the Early Cretaceous of China sheds light on the evolution of early ecological and cranial diversity in birds"

_PeerJ, doi:10.7717/peerj.1765_

## Round 0.1 · original submission · Major Revisions

Two reviewers have given thorough reviews to the MS, with some constructive criticisms. There are some serious concerns about the MS that mean that it is a long way still from being acceptable - I agree that it must be better illustrated and the VERY important issue of confidently establishing (or truly testing) the authenticity of all of the material is mandatory before we might be able to accept the paper. It will need to be re-reviewed by the same reviewers (ideally) to see if they are more convinced. I agree that more preparation would be wise and perhaps mandatory unless there is a convincing reason that this is impossible (lack of staff time would not be very convincing). If you choose to resubmit, ensure that you include a point-by-point Response to all of the reviewers' comments, please.

Reviewer 1 ·

Basic reporting

The article conforms to the requirements.

Experimental design

The article conforms to the requirements.

Validity of the findings

The paper represents a description of a new early cretaceous ornithurine bird with an unusual cranial morphology. This find provides an important contribution to our knowledge of the early evolution of skull in ornithurine birds, and, I believe, is of great interest for evolutionary ornithologists. However, many morphological details described in the paper are not clearly visible on the figures; in many cases I could not confirm the presence of the reported morphologies, and I think that some readers will face the same problem. The state of preservation of the specimen in some cases may allow various interpretations (at least this appears from the figures), and I believe that authors should provide more justification for their interpretations. Interpretative drawings of at least some parts of the specimen which will show bone margins and labeled bony structures will definitively help. I suggest that authors could add interpretative drawings of the skull, girdles (especially pectoral) and autopod regions, and label most discernable morphological details on them.
Importantly, another longirostrine bird has been described recently, Xinghaiornis, which looks similar to the specimen described in this paper. The two taxa definitively should be compared. Reference: Xuri, W., Chiappe, L.M., Fangfang, T., Qiang, J.I., Xinghaiornis lini (Aves: Ornithothoraces) from the Early Cretaceous of Liaoning: An Example of Evolutionary Mosaic in Early Birds, Acta Geologica Sinica - English Edition,2013, vol. 87, no. 3. pp. 686-689.


Minor points
Lines 17 and 18. The word “comparatively” is used twice in one sentence.
Line 17. “comparatively-close relatives of crown clade Aves” seems too fuzzy definition.
Line 31. Do you mean ecology not previously reported in early (basal; early Cretaceous) Ornithurae?
Line 65. Reference needed for terrestrial niches.
Line 67. “For these taxa”. Not clear, basal onithuromorpha or ornithurines?
Line 126. I would not say that the rostrum is extremely elongate (see curlews, for example).
Line 127. Would be good to tell how the length of the rostrum is measured.
Line 129. Insert space after Shanweiniao.
Line 130. Scolopacidae (family) have very different rostrums. Do you mean Scolopax?
Lines 130-146. Please show the described features on the figures (drawings?).
Lines 162-165. The posterior mandible is said to be crushed, but the dentary is said to be forked posteriorly. Is it discernable instead of crushing?
Lines 189-240. Please show described features.
Line 203. What is meant under the term “ventral process”? If it’s tuberculum ventrale than it’s usually not visible in cranial view.
Line 221. Please indicate, right or left femur. Here and for other elements throughout the description.
Lines 221, 228. The femur is “notably shorter than the tarsometatarsus” while the tarsometatarsus is “slightly more elongate” than the femur.
Line 223. Are the collateral ligaments into the hip joints? Mistake?
Line 261. Misspelling.
Line 289. It seems to me that the sentence about Gansus must be revised.
Line 345. Ornithurae – better basal Ornithurae , because Aves are also Ornithurae.
Line 369. Please label area of contact between frontals and premaxillae.
Line 373. Something is missing after “in”.
Line 559. Newly referred holotype sounds odd. Holotype is always new for an taxon. And holotype may not be referred to a taxon, because a taxon is established based on holotype.

Annotated reviews are not available for download in order to protect the identity of reviewers who chose to remain anonymous.

Reviewer 2 ·

Basic reporting

See general comments

Experimental design

See general comments

Validity of the findings

See general comments

Additional comments

The authors need to do a better job trying to convince readers that the specimen has not been artificially enhanced, given the decoloration of the central portion of the rostrum. The specimen needs serious preparation and given that the primary goal of this paper is to describe a new taxon (and new morphologies), the paper should have many more illustrations.

The introduction needs to be carefully checked for inaccuracies, and some confusing statements need to be edited. Here are some suggestions and guidance:

l. 38. The introduction of the authors’ Ornithurae clade as characterized by taxa having a short bony tail homologous with that of extant birds is confusing. The reduction of the tail (i.e., having a short bony tail) of birds has consistently been interpreted as a synapomorphy of a much basal clade (including early birds such as Confuciusornis, Sapeornis, and others), thus it is not the best attribute to characterize the clade Huang et al are referring to.

l.49. At least Iteravis huchzermeyeri 
and Gansus zheni are also known by multiple specimens. The number mentioned here should be emended to six.
l. 50. “The majority of the Chinese specimens of Early Cretaceous belong to one small-bodied and 
apparently terrestrial clade, Hongshanornithidae (Wang et al., 2015). “ This statement is not accurate. Many of the taxa mentioned in this first paragraph are clearly members of clades different from the Hongshanornithidae, as it is expressed in the cladogram of Figure 4.
l. 55. Dates for the Yixian Fm are typically around 125Ma not 129.7 and the reference is Chang et al. 2009 (not 2019).
l.56. Taxa other than Archaeorhynchus and Longicrusavis are known from the Yixian Formation – the authors should look carefully at the original literature.
l. 58. I believe the correct spelling of the formation containing the remains of Gansus yumenensis is ‘Xiagou’ not ‘Xiago’.
l. 62. Use ‘habitat’ instead of ‘habitus’
l. 66. Herbivorous and granivorous are partially overlapping diets: an animal that eats grains can be more specifically described as granivorous but also as a herbivorous animal.
Diagnosis. The Diagnosis primarily compares the new specimen with taxa within the crown clade, Hesperornis and Ichthyornis, both of which are obviously not the same as the new specimen. Yet, one wonders about comparisons to other Jehol birds—it seems that the authors simply rely on the length of the rostrum of the new specimen to diagnose a new clade. I believe they can do a better job.
Discussion. The paper is confusing, with self-contradicting statements here and there:
In the third paragraph of the Discussion, the authors state that “All known Mesozoic birds are characterized by plesiomorphic bill morphologies not seen in extant birds; the premaxilla in Aves is restricted to the tip of the facial margin while the maxilla makes up most of that margin.” It is confusing what the authors want to say in this statement: Aves includes all extant birds, so the condition within this clade encompasses rostra with both dominant maxilla and dominant premaxilla. Alternatively, if the authors use Aves to refer to the crown clade (as Clarke has used in many instances), then the statement incorrectly refers to extant birds having a rostrum in which the maxilla makes up most of the facial margin.
Furthermore, it is hard to imagine how such a poorly preserved and poorly prepared specimen can say anything about cranial kinesis—the authors need to have the specimen prepared adequately to be able to distinguish morphologies related to kinesis.
Lastly, Julia Clarke should take the time to edit the manuscript. It is plagued with typos and poorly written statements. Some examples:
l. 120. (diagnosis): public symphysis—I suppose the authors are referring to the pubic symphysis.
l. 128 ‘long-rostrumed’ Jehol taxa
l. 128. “longpteryid enantironithines” instead of longipterygid enantiornithine
l. 135. ‘enantiornithines with long-rostri’ – it should be with long-rostra.
l. 390 “So far it appears … as such should be further investigates as 
a possible a key innovation in their remarkable radiation.” 
- perhaps …as such should be further investigated as a possible key innovation…

---

## Round 0.2 · Minor Revisions

One reviewer has checked the MS and suggested some relatively minor changes to consider a recent publication, which if done in a reasonable fashion will bring the paper to a very acceptable standard and not require further review. Please resubmit the revised MS when ready and I will check it.

Reviewer 1 ·

Basic reporting

The presentation of the paper has been substantionally improved.

Experimental design

The presentation of the paper has been substantionally improved.

Validity of the findings

See Geeneral Comments section.

Additional comments

The authors have improoved the paper, and added better resolution close-up pictures that are very helpful. I appreciate that but I still cannot see many details, for example, the procoracoid process of the coracoid which is described but not labeled. But much more importantly, a new longirostrine taxon has been described recently (Dingavis, Journal of Systematic Paleontology) which also represents an ornithurine bird with the maxilla-based elongate rostrum. That paper, already published online, contribute much to the same evolutionary topic as the presented manuscript. Hence the presented manuscript must be revised in the light of what has been recently published.

---

## Round 0.3 · accepted · Accept

Thanks- I am convinced that the revised MS is entirely acceptable!

One tiny change on p3: ". While if new data shows that Xinghaiornis, Juehuaornis and Dingavis form a clade are the same genus, the genus name Juehuaornis would have priority for Dingavis longimaxilla and Changzuiornis ahgmi" needs to read "form a clade that constitutes the same genus" or similar.

and p9 " The fact that the elongate maxillae was not present in more derived taxa implies that in Aves this skull configuration provided less structural stability (O’Connor et al., 2016)."== "elongate maxillae were not present"